# The burden of *T. solium* cysticercosis and selected neuropsychiatric disorders in Mocuba district, Zambézia province, Mozambique

**Irene Langa**[1,2], **Fernando Padama**[3], **Noémia Nhancupe**[1,2], **Alberto Pondja**[4], **Delfina Hlashwayo**[5]*, **Lidia Gouveia**[6], **Dominik Stelzle**[7,8], **Clarissa Prazeres da Costa**[9], **Veronika Schmidt**[7,10], **Andrea S. Winkler**[7,10], **Emília Virgínia Noormahomed**[1,2,11]

**1** Microbiology Department, Parasitology Laboratory, Faculty of Medicine, Eduardo Mondlane University, Maputo, Mozambique, **2** Mozambique Institute of Health Education and Research (MIHER), Maputo, Mozambique, **3** Zambézia Operational Research Unit, Provincial Directorate of Health, Quelimane, Mozambique, **4** Faculty of Veterinary, Eduardo Mondlane University, Maputo, Mozambique, **5** Faculty of Sciences, Eduardo Mondlane University, Maputo, Mozambique, **6** Mental health Department; National Public Health Directorate, Ministry of Health Maputo, Maputo, Mozambique, **7** Center for Global Health, Department of Neurology, School of Medicine, Technical University of Munich, Munich, Germany, **8** Chair of Epidemiology, Department of Sport and Health Sciences, Technical University of Munich, Munich, Germany, **9** Institute for Medical Microbiology, Immunology and Hygiene, Center for Global Health, School of Medicine, Technical University of Munich, Munich, Germany, **10** Centre for Global Health, Institute of Health and Society, University of Oslo, Oslo, Norway, **11** Department of Medicine, Infectious Diseases Division, University of California, San Diego, California, United States of America

* delfinahlashwayo@gmail.com

## Abstract

### Background

*Taenia solium* (neuro-)cysticercosis, a neglected tropical disease, can be associated with epileptic seizures and other neuropsychiatric (= neurological and psychiatric) disorders. This study aimed to evaluate the association of *T. solium* cysticercosis with selected neuropsychiatric disorders and/or symptoms (chronic headache, epileptic seizures/epilepsy and psychosis) in Mocuba district, Mozambique.

### Methodology

Between March and May 2018, a cross-sectional study was conducted among 1,086 participants aged 2 years or above in Mocuba district, Zambézia province, central Mozambique, to assess the seroprevalence of human cysticercosis and risk factors for infection, as well as to explore its relation to selected neuropsychiatric disorders. Socio-demographic and clinical data were collected from each participant using a modified questionnaire designed by the Cysticercosis Working Group for Eastern and Southern Africa. Additionally, neuropsychiatric disorders, such as chronic headache, epileptic seizures/epilepsy and psychosis were assessed using four vignettes. *T. solium* antigen and cysticercosis IgG in serum were detected using both *T. solium* antigen B158/B60 enzyme linked immunosorbent assay (ELISA) and LDBIO Cysticercosis Western Blot, respectively.

**Data Availability Statement:** The data set is deposited in Zenodo repository and can be

accessed via the following link: https://doi.org/10.5281/zenodo.6815437.

**Funding:** The main funder for this study was the German Federal Ministry of Education and Research under CYSTINET-Africa project number 01KA1618 (EVN and ASW). Further support was received from the Fogarty International Center (FIC) - National Institute of Health (NIH) grant R25TW011216 (EVN) for training health professionals on the diagnosis and treatment of epilepsy. The funders had no role in study design, data collection and analysis, decision to publish, or preparation of the manuscript.

**Competing interests:** The authors have declared that no competing interests exist.

## Principal findings

Overall, 112/1,086 participants (10.3%) were sero-positive for *T. solium* antigen or antibodies. Prevalence of antibodies (6.6%; n = 72) was higher than of antigens (4.9%; n = 54). In the questionnaires, 530 (49.5%) of participants reported chronic headache, 293 (27%) had generalized epileptic seizures, 188 (18%) focal seizures and 183 (18.3%) psychosis. We found a statistically significant association between seropositivity for *T. solium* and chronic headache (p = 0.013). Additionally, increasing age (p = 0.03) was associated with Ag-ELISA seropositivity.

## Conclusions

Our study revealed that in Mocuba, *T. solium* cysticercosis is prevalent and associated with self-reported chronic headache. Additionally, in the study setting, the seroprevalence of cysticercosis increased with age. However, it is not associated with other neuropsychiatric disorders such epileptic seizures/epilepsy and psychosis. Future studies are needed to confirm the high burden of neuropsychiatric disorders and their possible etiology, including neurocysticercosis, using additional serological, molecular biological and radiological diagnostic tools, as well as in-depth clinical examinations.

## Author summary

*Taenia solium* cysticercosis (TSC) is a neglected tropical disease caused by the larval stage of *T. solium*. The disease is a serious threat to public health, especially in low-income countries and is associated with poor pig husbandry practices, deficient hygiene and sanitation, close contact between humans and pigs, lack of slaughter facilities for pigs and inadequate meat inspection. When the larva of *T. solium* lodges in the human central nervous system, the disease is called neurocysticercosis (NCC). The most frequent neurological manifestations of NCC are epileptic seizures and epilepsy. NCC diagnosis remains a challenge in low-income countries and its relationship with neurological signs/symptoms so far was not studied in Mocuba, located in Zambézia province, one of the districts with the biggest pig populations in Mozambique. In this study we investigated the seroprevalence for TSC, risk factors for infection and the association with neuropsychiatric (= neurological and psychiatric) disorders. Seroprevalence of TSC was 10.3%, and was associated with chronic headache. Moreover, increasing age was associated with Ag-ELISA seropositivity. Future studies are needed applying in-depth clinical examination, and additional serological, molecular biological and radiological diagnostic tools in order to confirm (or not) the results of our study.

## Introduction

*Taenia solium* cysticercosis (TSC) is a foodborne, zoonotic and neglected tropical disease (NTD), caused by the larval stage (cysticercus) of *T. solium*. The disease is emerging as a serious threat to public health, inflicting significant economic losses and disabilities in low-income countries of sub-Saharan Africa, Latin America and Asia where *T. solium* is endemic. It is also

of increasing concern in non-endemic countries due to globalization and the immigration of tapeworm carriers [1–5].

Humans are the definitive host and thus carriers of the adult tapeworm which develops after ingestion of measly raw or undercooked pork. However, humans may also become intermediate hosts and develop TSC following direct or indirect ingestion of *T. solium* eggs which develop into cystic larvae in various parts of the body [1,5–8]. When the larva lodges in the central nervous system (CNS), the disease is called neurocysticercosis (NCC). The most frequent neurological manifestations include epileptic seizures and epilepsy. In endemic areas, up to 30% of people with epilepsy may suffer from NCC [1–5,9]. Other neurological signs and symptoms include severe headache, intracranial hypertension, dementia, blindness, or chronic meningitis. Also, different neuropsychiatric disorders have been associated with NCC although there is only limited data available [1,5,7,8,10].

NCC prevails due to the maintenance of *T. solium* life cycle, associated with poor pig husbandry practices, deficient hygiene and sanitation, close contact between humans and pigs, the intermediate host, lack of slaughter facilities for pigs and inadequate meat inspection [1,7,8,11,12]. However, the diagnosis of NCC remains a challenge because of the poor specificity of clinical and neuroimaging findings, the latter including magnetic resonance imaging (MRI) and computed tomography (CT scan), as well as suboptimal predictive values in immunodiagnostic tests, particularly in endemic settings [1,2].

The few available cross-sectional serological studies from Mozambique found that the seroprevalence of either antibodies or antigens against *T. solium* cysticercerci varied between 15% to 21% in apparently healthy children and adults, while in neurological patients the seroprevalence of cysticercosis can be as high as 51% [13–15]. Further, the only published community-based study conducted in Angónia district, in the central western part of the country, found that amongst 151(8.8%) people confirmed to be living with epilepsy, 107 (70.9%) were seropositive to cysticercus antigens [14]. These data suggest that NCC might be potentially one of the main causes of epileptic seizures/epilepsy and other neurological and/or psychiatric disorders in Mozambique.

Therefore, this study investigated the seroprevalence for *T. solium* cysticercosis and its associated risk factors as well as neuropsychiatric (= neurological and psychiatric) disorders with an emphasis on chronic headache, epileptic seizures/epilepsy and psychosis in both children and adults from Mocuba district, Zambézia province, Mozambique.

## Methods

### Ethics statement

The present study was approved by the Mozambique National Bioethical Committee of Health (51/CNBS/2017) and by the administrative authorities from Zambézia province and community leaders. The study also received approval from the ethics committee of the Klinikum rechts der Isar, Technical University of Munich, Germany (537/18 S-KK).

Prior to the interviews and blood sampling, written consent were obtained from the enrolled participants. Written consent from children aged below 12 years and illiterate participants were obtained in the presence of a guardian or witness, who signed on their behalf. For participants aged between 12 and 17 years a written assent was obtained in addition to their parents' consent. Each participant was assigned a unique identification study number and all data and samples were handled confidentially using this study number from this point onward.

Participants who suffered from epileptic seizures/epilepsy or other neuropsychiatric disorders were referred to the local health centre for diagnosis confirmation and follow-up according to the Mozambique Ministry of Health guidelines [16].

### Study area and population

The study was conducted in seven villages of Mocuba district (Fig 1) located in Zambézia province, central Mozambique (16˚50' 13" S, 36˚59' 14" E). The district has an area of 8,803 km$^2$ and an altitude of 200–400 meters above sea level.

The total population is estimated at 422,681. There is an illiteracy rate of 62.5% with most of the population (82.6%) living in rural areas. Moreover, the disease profile of Mocuba is dominated by infectious diseases including many NTDs with higher prevalence estimates. As

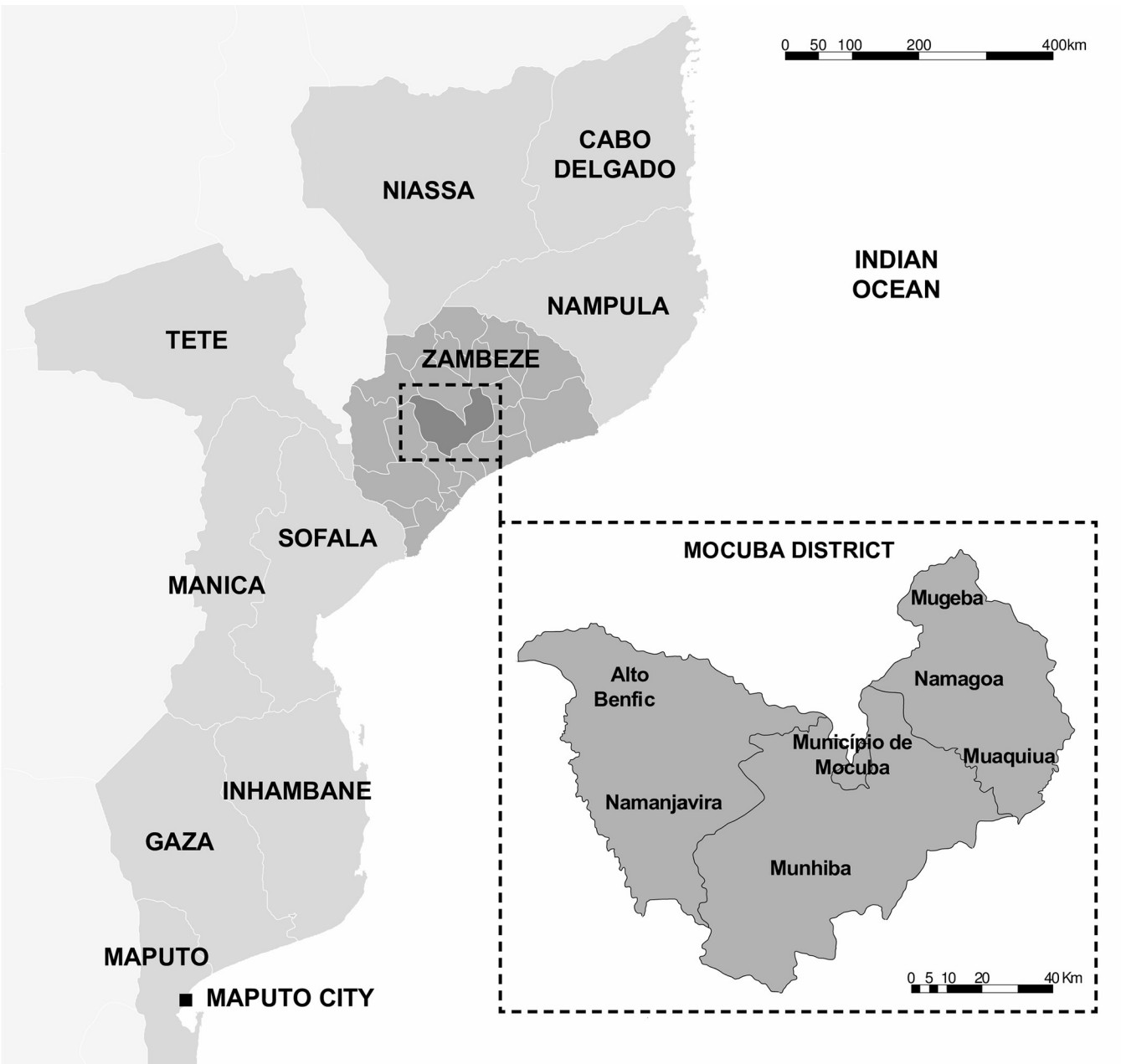

**Fig 1. Location of the study area.** Created by Patricia Noormahomed based on EarthExplorer (https://earthexplorer.usgs.gov).

an example, the district has an HIV prevalence of 9% and 39% of children are infected by at least one intestinal parasite [17–21].

## Study design and sampling

A community-based cross-sectional study was conducted between March and May 2018. The sampling frame was constructed using information of the National Institute of Statistics [22]. For the sample size calculation, it was estimated that 15% of households would have at least one member with cysticercosis based on a study conducted in a northwestern district of Mozambique [14]. Assuming a 95% confidence level and 20% precision, and correcting for a finite population of households in the study area [22], a sample size of at least 543 households was obtained by using the formula $n = Z^2p(1—p)/d^2$ [23], where *n* is the sample size required, *Z* is the *z*–score for the desired confidence level, *p* is the expected proportion of households, and *d* is the desired precision relative to the expected proportion of households. The sampling procedure followed a two-stage household-based design, whereby Primary Sampling Units (PSUs) consisted of villages, and households were chosen during the second stage. Prior to selection of households, the village authorities were approached and the purpose of the study was explained. Once the permission was granted, a systematic random sampling with probability proportional to size (PPS) technique was used to select households. Two eligible participants from each household were recruited randomly into the study (Fig 2). When only one

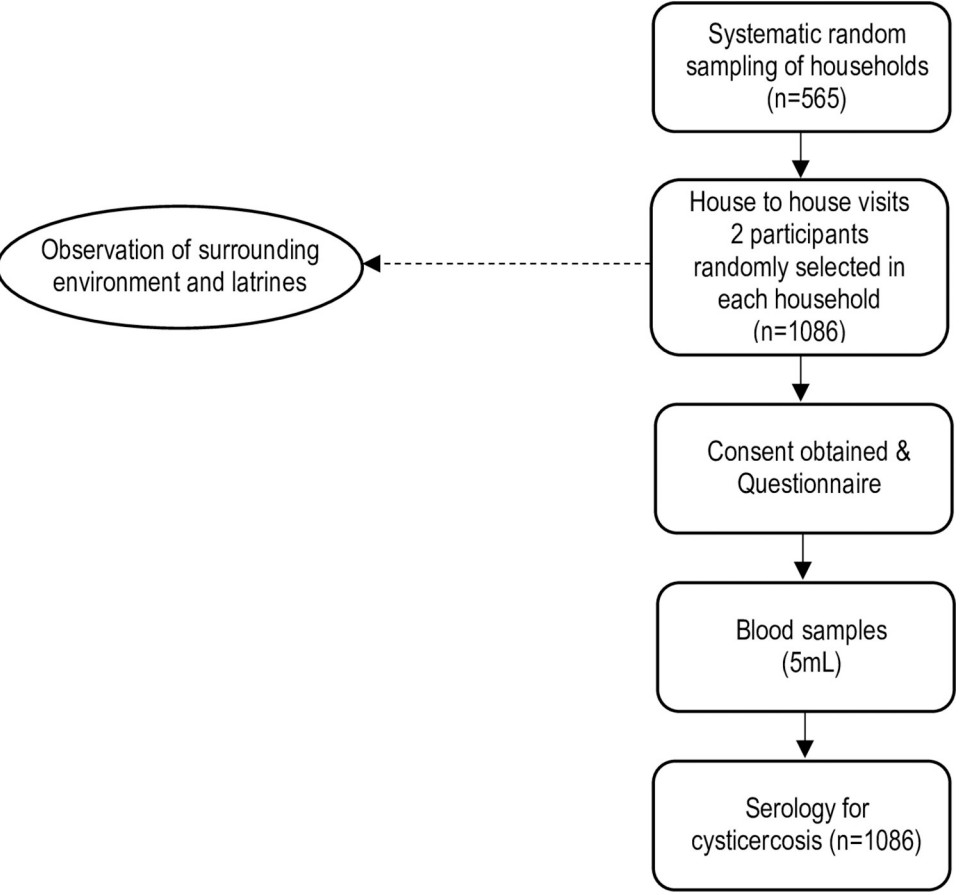

**Fig 2. Data collection workflow of the study sample.**

member was available in a selected household, either by absence or refusals of other members, participants were recruited from the nearest household. Inclusion criteria were living in the household and being at least 2 years old. People who had taken alcoholic beverages in the previous 24 hours were excluded. In total, 1,086 participants from 565 households were included in the study.

## Questionnaire survey and vignettes

A pre-designed questionnaire developed by the Cysticercosis Working Group for Eastern and Southern Africa (CWGESA) was used, with some modifications, that included four vignettes on neuropsychiatric disorders [24,25]. The four vignettes included chronic headache, epileptic seizures/epilepsy (generalized and focal) and psychotic disorders, and were confirmed with a neurologist (ASW). This was the major tool used to establish a diagnosis of neuropsychiatric (= neurological and psychiatric) disorders in the study. The questionnaire also covered information on socio-demographic and risk factors associated with cysticercosis such as age, gender, education, sources of drinking water, use of latrines, rearing pigs and pig management practices, consumption of pork, awareness about TSC transmission, risk-related habits and practices.

Furthermore, direct observation to visualize the existence of latrines and usage and pig husbandry system in the household was made by the enumerators. The questionnaire designed in Portuguese language was used for the study, translated to the local language (Lomué) and back translated to Portuguese.

The survey was carried out by locally selected enumerators with knowledge of the local language, experience with community surveys, and basic training on TSC transmission, prevention and clinical signs/symptoms of selected neuropsychiatric disorders. The training of the enumerators was done via workshop sessions and pre-tested prior to commencement of the study.

## Vignette 1. Assessment of chronic headache

To assess chronic headache, participants were asked if they had a history of daily headache that occurred for at least 15 consecutive days and for longer than three months. We did not ask for severity or any specific characteristics of headaches in order to keep it more general and allow for secondary causes of chronic headache such as NCC.

## Vignette 2. Assessment of generalized tonic-clonic seizures

To assess generalized tonic-clonic seizures (GTCS), we asked the participant if he/she had ever experienced a sudden loss of consciousness that caused him/her to fall to the ground or whether he/she was told about such an event by someone else. We also asked if the eyes were turned upwards, the body became rigid and/or involuntary violent movements of the arms and legs were observed. In addition, we asked if they were told or noticed that they were drooling and/or urinating and/or biting their tongue. We also asked if the person was asleep or slowed down or even confused after the actual epileptic seizure and could not remember what happened.

## Vignette 3. Assessment of focal seizures

To assess focal seizures (FS), participants were asked whether they had ever noticed or been told about uncontrolled movements in body parts such as an arm or leg without loss of

consciousness or before loss of consciousness. For the latter, we also asked about warning signs such as a strange smell (aura) before the epileptic seizure.

### Vignette 4. Assessment of psychotic disorders

To assess psychotic behaviour and/or hallucinations, we asked the participant if he/she was informed of any changes in behaviour, such as becoming very unstable, appearing anxious and/or doing inappropriate things, such as undressing or harming others for no reason. They were also asked if they heard strange voices that no one else could hear, or if they saw things that were not real to others. They were also asked if they had the impression of being followed.

### Blood sampling

A 5 ml sample of venous blood was obtained by venepuncture from each of the recruited individuals after consenting [26]. The samples were immediately stored in cooling boxes at 4˚C and transported to the Mocuba district Hospital Laboratory, where they were centrifuged for 5 min at 1500 r.p.m. to obtain serum. Serum was stored at -20˚C until shipment to the Parasitology Laboratory at Faculty of Medicine, Eduardo Mondlane University (UEM) in Maputo where samples were stored at -80˚C until further processing.

### Serological testing

*T. solium* circulating antigens (Ag) were detected using the monoclonal antibody based B158C11A10 Enzyme Linked Immunosorbent Assay (Cysticercosis Ag-ELISA, ApDia, Belgium). Ag index was calculated as the mean of optical density (OD) of each serum sample divided by the cut-off. In addition, IgG antibodies (Ab) against *T. solium* larva were determined via *Western Blot* (LDBIO Diagnostic, Lyon, France). A sample was considered positive if at least 2 bands (6–8 kDa; 12 kDa; 23–26 kDa; 39 kDa; 50–55 kDa) were visible, according to the manufacturer's instructions.

### Statistical analysis

Data were captured in EpiData Entry Client version 4.3 and analysed using STATA version 13.0 (Stata Corp., College Station, TX). Descriptive analyses were first performed by calculating frequencies and percentages for categorical variables. Age was categorized into age groups (2–14, 15–24, 25–54 and >55 years) for the purpose of analysis. Using Chi-square test, bivariate analysis was performed to assess associations between a positive Ag-ELISA and/or *Western Blot* assay results and several factors. These factors included: socio-demographic and neuropsychiatric diseases; knowledge about *T. solium*; transmission risk-related variables and pig management variables. Chi-square tests were also used to assess difference in seroprevalence by district. Logistic regression models were performed to assess the association of seroprevalence of *T. solium* with the four vignettes. Models were run unadjusted and adjusted for age, gender, education and occupation. The significance level was set at p<0.05.

### Results

Among the 1,086 participants included in the study, the age ranged from 2 to 87 years, with a median age of 25 years. A total of 656 participants (60.4%) were predominantly farmers, 640 (58.9%) were female, and the majority 763 (70.3%) attended primary school. A total of 237 (21.8%) participants reported outdoor defecation, 806 (74.2%) reported eating pork, 155 (14.3%) were pig keepers and 105 (9.7%) have heard about the pork tapeworm (Table 1).

The results of cysticercosis serological screening either for antigen or antibody detection or both from each village are shown in Fig 3. Overall, out of the 1,086 samples tested, 112 (10.3%) were positive in at least one serological test; 54 (4.9%) had circulating antigens and 72 (6.6%) had circulating antibodies to *T. solium* larva. Moreover, 14 (1.3%) were positive in both tests. A wide variation of seropositivity was observed across different villages ranging from 3.6% in Alto Benfica village to 17.7% in Mocuba Municipality (p = 0.007).

**Table 1. Socio-demographic characteristics of the study participants.**

| Variables | Study participants | |
|---|---|---|
| | **n** | **(%)** |
| **Age group (years)** | | |
| 2–14 | 277 | 25.5 |
| 15–24 | 251 | 23.1 |
| 25–54 | 457 | 42.1 |
| ≥ 55 | 101 | 9.3 |
| **Gender** | | |
| Male | 446 | 41.1 |
| Female | 640 | 58.9 |
| **Level of education** | | |
| No formal education | 197 | 18.1 |
| Primary | 763 | 70.3 |
| Secondary or higher | 121 | 11.1 |
| Other | 5 | 0.5 |
| **Occupation** | | |
| Farmer | 656 | 60.4 |
| Trading | 28 | 2.6 |
| Student | 245 | 22.6 |
| State employee | 17 | 1.6 |
| Other | 140 | 12.9 |
| **Water source** | | |
| River | 82 | 7.6 |
| Well | 585 | 53.9 |
| Borehole | 405 | 37.3 |
| Tap | 14 | 1.2 |
| **Faecal disposal** | | |
| Latrine | 849 | 78.2 |
| Outdoor defecation | 237 | 21.8 |
| **Pork consumption** | | |
| Yes | 806 | 74.2 |
| No | 280 | 25.8 |
| **Pork preparation** | | |
| Boiled | 452 | 56.1 |
| Grilled | 345 | 42.8 |
| Other | 9 | 0.1 |
| **Pig keeping** | | |
| Yes | 155 | 14.3 |
| No | 931 | 85.7 |
| **Ever heard about tapeworm** | | |
| Yes | 105 | 9.7 |
| No | 981 | 90.3 |

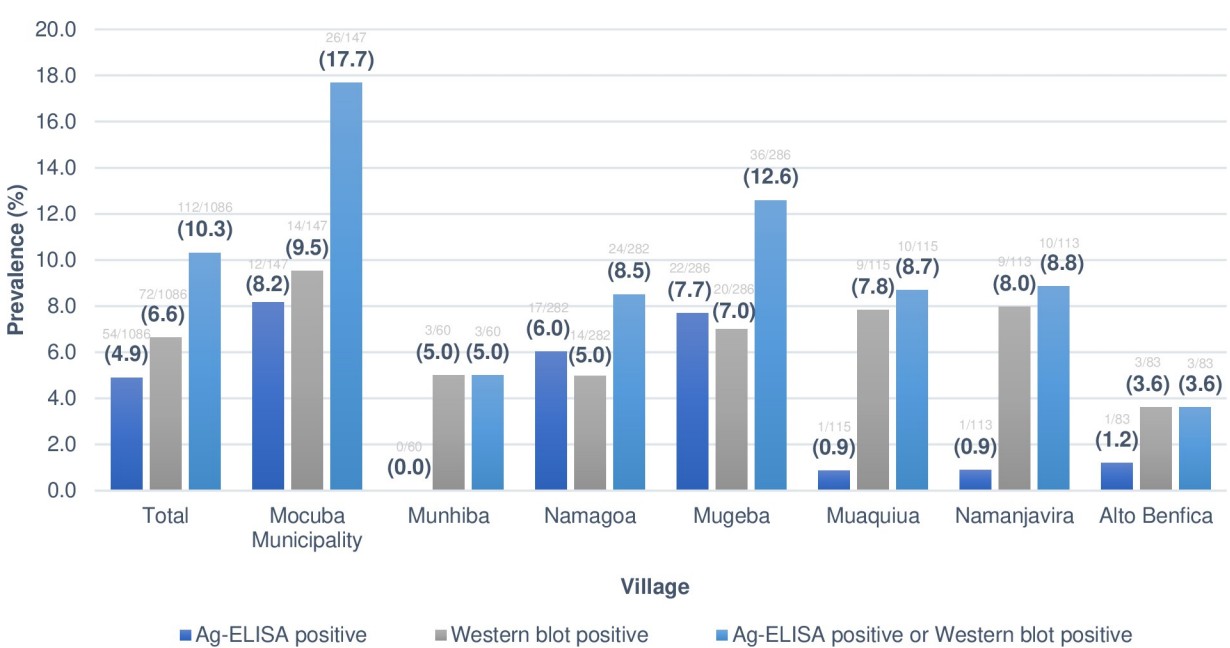

**Fig 3. Prevalence of human cysticercosis according to each study village in Mocuba district.**

In general, the seroprevalence of Ab tended to be higher (ranging from 3.6% in Alto Benfica village to 9.5% in Mocuba Municipality) than that of Ag (ranging from 0% in Munhiba to 8.2% in Mocuba Municipality; p<0.005). The details about serological assay results from each of the studied villages can be seen in Fig 3.

The analysis of socio-demographic variables in relation to the cysticercosis serological assays are presented in Table 2. Seropositivity for antigen or antibody increased with age and was lowest for children under the age of 15 years (p = 0.03). Seropositivity was not statistically significantly different between males and females (11.7% versus 9.4%, p = 0.22), nor between the levels of education (p = 0.12).

The findings concerning the assessment of clinical signs and symptoms related to neuro-psychiatric disorders in relation to socio-demographic factors are presented in Table 3. A total of 530 (49.5%) participants screened positive for vignette 1. More males screened positive than females (p = 0.02). Education was not associated with screening positive but farmers more commonly screened positive than people with other occupations (p = 0.05). Overall, 293 participants (27%) screened positive for vignette 2; positivity decreased with increasing age and was similar for males and females. Vignette 3 was positive in 188 (18%) of the participants and vignette 4 in 183 (18.3%). Vignette 4 was more commonly positive among participants older than 14 years and was more commonly positive among farmers. Of the total vignettes, chronic headache was associated with cysticercosis seropositivity (p = 0.013) (Table 4).

## Discussion

Our results indicate that *T. solium* antigens and antibodies were present in 10.3% of the study participants from Mocuba district. Moreover, chronic headache presented a significant association with cysticercosis seropositivity (p = 0.013), although neuropsychiatric disorders such as epileptic seizures/epilepsy and psychosis presented no significant association. These findings suggest that TSC is prevalent in Mocuba and chronic headache prevails as an important

**Table 2. Socio-demographic data and relation with serological assays to *T. solium* larva antigens and antibodies.**

| Variables | Study participants n (%) | Ag-Elisa [+] | Western blot [+] | Ag-Elisa [+] and Western blot [+] | Ag-Elisa [+] or Western blot [+] | |
|---|---|---|---|---|---|---|
| | | n (%) | n (%) | n (%) | n (%) | *p*-value |
| **Total** | 1086 | 54 (4.9) | 72 (6.6) | 14 (1.3) | 112 (10.3) | |
| **Age group (years)** | | | | | | |
| 2–14 | 277 | 4 (1.4) | 13 (4.7) | 1 (0.4) | 16 (5.8) | 0.03 |
| 15–24 | 251 | 12 (4.8) | 17 (6.8) | 1 (0.4) | 28 (11.2) | |
| 25–54 | 457 | 30 (6.6) | 37 (8.1) | 10 (2.2) | 57 (12.5) | |
| ≥55 | 101 | 8 (7.9) | 5 (5.0) | 2 (2) | 11 (10.9) | |
| **Gender** | | | | | | |
| Female | 640 | 26 (4.1) | 42 (6.6) | 6 (1.3) | 60 (9.4) | 0.22 |
| Male | 446 | 28 (6.3) | 30 (6.7) | 8 (1.3) | 52 (11.7) | |
| **Level of education** | | | | | | |
| No formal education | 197 | 5 (2.5) | 7 (3.6) | 0 (0) | 12 (6.1) | 0.12 |
| Primary | 763 | 40 (5.2) | 55 (7.2) | 14 (1.8) | 81 (10.6) | |
| Secondary or higher | 121 | 9 (7.4) | 10 (8.3) | 0 (0) | 19 (15.7) | |
| Other | 5 | 0 (0.0) | 0 (0.0) | 0 (0) | 0 (0.0) | |
| **Occupation** | | | | | | |
| Farmer | 656 | 38 (5.8) | 43 (6.6) | 11 (1.7) | 70 (10.7) | 0.63 |
| Other | 430 | 16 (3.7) | 29 (6.7) | 3 (0.7) | 42 (9.8) | |
| **Water source** | | | | | | |
| River | 82 | 7 (8.5) | 5 (6.1) | 3 (3.7) | 9 (11.0) | 0.67 |
| Well | 585 | 28 (4.8) | 43 (7.4) | 6 (1) | 65 (11.1) | |
| Borehole | 405 | 19 (4.7) | 22 (5.4) | 5 (1.2) | 36 (8.9) | |
| Tap | 14 | 0 (0.0) | 2 (14.3) | 0 (0) | 2 (15.4) | |
| **Faecal disposal** | | | | | | |
| Latrine | 849 | 47 (5.5) | 56 (6.6) | 13 (1.5) | 90 (10.6) | 0.56 |
| Outdoor defecation | 237 | 7 (3.0) | 16 (6.8) | 1 (0.4) | 22 (9.3) | |
| **Pork consumption** | | | | | | |
| Yes | 806 | 43 (5.3) | 57 (7.1) | 3 (1.1) | 89 (11.0) | 0.18 |
| No | 280 | 11 (3.9) | 15 (5.4) | 11 (1.4) | 23 (8.2) | |
| **Pork preparation** | | | | | | |
| Boiled | 452 | 21 (4.6) | 29 (6.4) | 7 (1.5) | 43 (9.5) | 0.13 |
| Grilled | 345 | 22 (6.4) | 28 (8.1) | 4 (1.2) | 46 (13.3) | |
| Not specified | 9 | 0 (0.0) | 0 (0.0) | 0 (0) | 0 (0.0) | |
| **Source of pork** | | | | | | |
| Informal market | 747 | 41 (5.5) | 52 (7.0) | 10 (1.3) | 83 (11.1) | 0.88 |
| Butchery | 2 | 0 (0.0) | 0 (0.0) | 0 (0) | 0 (0.0) | |
| Other | 56 | 2 (3.6) | 5 (8.9) | 1 (1.8) | 6 (10.7) | |
| **Pig keeping** | | | | | | |
| Yes | 155 | 10 (6.5) | 14 (9.3) | 3 (1.9) | 21 (13.5) | 0.15 |
| No | 931 | 44 (4.7) | 58 (6.2) | 11 (1.2) | 91 (9.8) | |
| **Pig husbandry system** | | | | | | |
| Confinement | 52 | 6 (11.5) | 4 (7.7) | 1 (1.9) | 9 (17.3) | 0.33 |
| Free ranging | 103 | 4 (3.9) | 10 (9.7) | 2 (1.9) | 12 (11.7) | |
| **Ever heard about the tapeworm** | | | | | | |
| Yes | 105 | 2 (1.9) | 6 (5.7) | 1 (1) | 7 (6.7) | 0.20 |
| No | 981 | 52 (5.3) | 66 (6.7) | 13 (1.3) | 105 (10.7) | |

**Table 3. Assessment of clinical signs and symptoms related to neuropsychiatric disorders in relation to socio-demographic factors.**

| Variables | Study participants n (%) | Vignette 1 (Chronic headache) | | Vignette 2 (Tonic clonic seizure) | | Vignette 3 (Focal seizure) | | Vignette 4 (Psychotic disorder) | |
|---|---|---|---|---|---|---|---|---|---|
| | | n (%) | *p*-value | n (%) | *p*-value | n (%) | *p*-value | n (%) | *p*-value |
| **Total** | 1086 | 530/1070 (49.5) | | 293/1086 (27) | | 188/1042 (18) | | 183/999 (18.3) | |
| **Age group (years)** | | | | | | | | | |
| 2–14 | 277 | 123/268 (45.9) | 0.47 | 89/277 (32.1) | <0.01 | 45/251 (17.9) | 0.55 | 23/219 (10.5) | <0.01 |
| 15–24 | 251 | 125/247 (50.6) | | 73/251 (29.1) | | 48/247 (19.4) | | 49/242 (20.2) | |
| 25–54 | 457 | 227/454 (50) | | 116/457 (25.4) | | 82/444 (18.5) | | 92/441 (20.9) | |
| ≥55 | 101 | 55/101 (54.5) | | 15/101 (14.9) | | 13/100 (13) | | 19/97 (19.6) | |
| **Gender** | | | | | | | | | |
| Female | 446 | 197/438 (45) | 0.02 | 128/446 (28.7) | 0.32 | 72/424 (17) | 0.51 | 84/407 (20.6) | 0.14 |
| Male | 640 | 333/632 (52.7) | | 165/640 (25.8) | | 116/618 (18.8) | | 99/592 (16.7) | |
| **Level of education** | | | | | | | | | |
| No formal education | 197 | 96/187 (51.3) | 0.74 | 60/197 (30.5) | 0.58 | 36/181 (19.9) | 0.90 | 23/157 (14.6) | 0.39 |
| Primary | 763 | 377/757 (49.8) | | 199/763 (26.1) | | 132/743 (17.8) | | 136/718 (18.9) | |
| Secondary or higher | 121 | 55/121 (45.5) | | 32/121 (26.4) | | 19/113 (16.8) | | 24/119 (20.2) | |
| Other | 5 | 2/5 (40) | | 2/5 (40) | | 1/5 (20) | | 0/5 (0) | |
| **Occupation** | | | | | | | | | |
| Farmer | 656 | 338/650 (52) | 0.05 | 162/656 (24.7) | 0.04 | 113/646 (17.5) | 0.61 | 130/635 (20.5) | 0.03 |
| Other | 430 | 192/420 (45.7) | | 131/430 (30.5) | | 75/396 (18.9) | | 53/364 (14.6) | |
| **Water source** | | | | | | | | | |
| River | 82 | 41/79 (51.9) | 0.80 | 23/82 (28) | 0.77 | 19/77 (24.7) | 0.24 | 12/73 (16.4) | 0.73 |
| Well | 585 | 292/582 (50.2) | | 165/585 (28.2) | | 102/560 (18.2) | | 104/531 (19.6) | |
| Borehole | 405 | 190/395 (48.1) | | 101/405 (24.9) | | 63/392 (16.1) | | 66/382 (17.3) | |
| Tap | 13 | 7/13 (53.8) | | 4/13 (30.8) | | 4/12 (33.3) | | 1/12 (8.3) | |
| Other | 1 | 0/1 (0) | | 0/1 (0) | | 0/1 (0) | | 0/1 (0) | |
| **Faecal disposal** | | | | | | | | | |
| Latrine | 820 | 396/810 (48.9) | 0.57 | 220/820 (26.8) | 0.88 | 138/784 (17.6) | 0.62 | 132/765 (17.3) | 0.13 |
| Outdoor defecation | 237 | 121/231 (52.4) | | 64/237 (27) | | 46/230 (20) | | 43/208 (20.7) | |
| Other | 29 | 13/29 (44.8) | | 9/29 (31) | | 4/28 (14.3) | | 8/26 (30.8) | |
| **Pork consumption** | | | | | | | | | |
| Yes | 280 | 127/276 (46) | 0.20 | 64/280 (22.9) | 0.08 | 40/269 (14.9) | 0.14 | 43/260 (16.5) | 0.44 |
| No | 806 | 403/794 (50.8) | | 229/806 (28.4) | | 148/773 (19.1) | | 140/739 (18.9) | |
| **Pork preparation** | | | | | | | | | |
| Boiled | 452 | 192/446 (43) | <0.001 | 113/452 (25) | 0.04 | 72/431 (16.7) | 0.15 | 62/411 (15.1) | <0.01 |
| Grilled | 345 | 208/341 (61) | | 112/345 (32.5) | | 74/333 (22.2) | | 77/320 (24.1) | |
| Not specified | 9 | 3/7 (42.9) | | 4/9 (44.4) | | 2/9 (22.2) | | 1/8 (12.5) | |
| **Source of pork** | | | | | | | | | |
| Informal market | 747 | 372/736 (50.5) | 0.36 | 211/747 (28.2) | 0.08 | 136/719 (18.9) | 0.49 | 128/688 (18.6) | 0.44 |
| Butchery | 2 | 2/2 (100) | | 2/2 (100) | | 1/2 (50) | | 0/2 (0) | |
| Other | 56 | 29/55 (52.7) | | 16/56 (28.6) | | 11/51 (21.6) | | 12/48 (25) | |
| **Pig keeping** | | | | | | | | | |
| Yes | 155 | 66/153 (43.1) | 0.11 | 44/155 (28.4) | 0.74 | 27/146 (18.5) | 0.97 | 31/146 (21.2) | 0.39 |
| No | 931 | 464/917 (50.6) | | 249/931 (26.7) | | 161/896 (18) | | 152/853 (17.8) | |
| **Pig husbandry system** | | | | | | | | | |
| Confinement | 52 | 21/52 (40.4) | 0.75 | 15/52 (28.8) | 1 | 9/49 (18.4) | 1 | 9/50 (18) | 0.63 |
| Free ranging | 103 | 45/101 (44.6) | | 29/103 (28.2) | | 18/97 (18.6) | | 22/96 (22.9) | |
| **Ever heard about the tapeworm** | | | | | | | | | |
| Yes | 105 | 47/103 (45.6) | 0.47 | 34/105 (32.4) | 0.23 | 26/101 (25.7) | 0.05 | 20/102 (19.6) | 0.83 |
| No | 981 | 483/967 (49.9) | | 259/981 (26.4) | | 162/941 (17.2) | | 163/897 (18.2) | |

**Table 4. Analysis of statistical association between screening positive for vignettes and cysticercosis serology positivity.**

| Variables | Study participants n (%) | Ag-ELISA+ or Western blot+ | | | | |
|---|---|---|---|---|---|---|
| | | n (%) | Unadjusted Odds Ratio (95% CI) | p-value | Adjusted Odds Ratio (95% CI) | p-value |
| **Total** | 1086 | 112 (10.3) | | | | |
| **Vignette 1 (Chronic headache)** | | | | | | |
| No | 540 | 69 (12.8) | Reference | 0.013 | Reference | 0.018 |
| Yes | 530 | 43 (8.1) | 0.6 (0.4–0.9) | | 0.61 (0.41–0.91) | |
| **Vignette 2 (Tonic clonic seizure)** | | | | | | |
| No | 793 | 85 (10.7) | Reference | 0.47 | Reference | 0.59 |
| Yes | 293 | 27 (9.2) | 0.85 (0.53–1.32) | | 0.88 (0.55–1.38) | |
| **Vignette 3 (Focal seizure)** | | | | | | |
| No | 854 | 89 (10.4) | Reference | 0.93 | Reference | 0.90 |
| Yes | 188 | 20 (10.6) | 1.02 (0.6–1.68) | | 1.04 (0.6–1.71) | |
| **Vignette 4 (Psychotic disorder)** | | | | | | |
| No | 816 | 87 (10.7) | Reference | 0.91 | Reference | 0.62 |
| Yes | 183 | 19 (10.4) | 0.97 (0.56–1.61) | | 0.88 (0.5–1.46) | |

Models were adjusted for age, gender, education and occupation

disorder in seropositive patients. To the best of our knowledge, this was the first pilot study performed in the country, particularly in Mocuba district with the aim to generate baseline information about seroprevalence and risk factors for human cysticercosis, and its possible relationship with some selected neuropsychiatric disorders. There were several important findings from our study.

## Serology

Our findings revealed that human cysticercosis was prevalent in the setting of our study (10.3%), although lower than previously reported in Angónia district and Beira city, Central Mozambique [14,15]. The prevalence is still worrying as it demonstrates the potential for a greater spread of the disease, which can be caused by several factors, including hygiene and sanitation. The higher prevalence in some municipalities such as Mocuba (17.7%) in comparison with those with lower prevalences (e.g. Alto Benfica, 3.6%), may be explained by sociodemographic factors, which should be analysed in future studies.

Previous studies from Mozambique found in general higher prevalence of either antigens (15%) or antibodies (10%) to *T. solium* larva than the ones reported in this study [14,15], and were consistent with studies done in other settings such as Cameroon [27], Nigeria [28], Vietnam [29,30], Zambia [31] and Democratic Republic of Congo [5], which reported similar seroprevalence rates. In our study, only 12.5% of the screened sera tested simultaneously positive to both serological assays. The differences in the prevalence of antigens (4.9%), that indicates a current infection, and antibodies (6.2%), which indicates a previous or current exposure to infection, may be influenced by the number, location of cysts and their larval stage. Furthermore, antibodies can persist longer, even after the clearance of the infection, while the presence of antigens is only detected when viable parasites are present [1–3,10,32,33]. Moreover, the western blot assay we used detected only three out of the seven glycoproteins described to be

specific for cysticercosis [1,32,34]. So, it is possible that the prevalence of antibodies found in our study was underestimated.

In addition to that, the differences in study design, HIV serostatus characteristics of the study participants (immunocompetent or immunocompromised), serological assays used for each study, and parasite genetic diversity (which ultimately influenced the sensitivity and specificity of the serological assays) can explain the differences found within studies in the same region and in other settings [5,9,10,35]. Of note, that to date, we do not have systematic studies done in Mozambique accessing the existing genetic variants of *T. solium*, except for the one in which phylogenetic studies of cysts from Central Mozambique revealed the existence of the Latin variant [32]. As Mozambique has been exposed to migration from Europe and Asia, we cannot exclude the possibility that Mozambique has both variants circulating in different regions and further studies should be done in order to provide relevant information about this subject.

## Cysticercosis serology and neuropsychiatric disorders

Our study also demonstrated that chronic headache (49.5%), generalized epileptic seizures (27%), focal epileptic seizures (18%), and psychosis (18.3%) affected a substantial proportion of study participants irrespective of their cysticercosis serostatus. Our findings are consistent with data from 2018 hospital records, indicating that Mocuba district had the highest number of patients (7,416; 75.3%) in the entire province (9,844) with mental disorders. Out of these 7,416 mental health out-patients, 1,863 (25.1%) had epilepsy, 250 (3.4%) had mental retardation and 1,401 (18.9%) had psychosis [36].

In our study, epileptic seizures were significantly more prevalent in younger participants (2–14 years) and decreased with age (p<0.001). This finding is consistent with recent mental health data in Mozambique showing that among people living with epilepsy, 54% were children aged up to 15 years, although the underlying causes of these conditions are not well defined [21,37]. Similar studies in African countries also found that epileptic seizures and epilepsy were the main leading causes of mental health diagnoses in out-patients' consultations and that the majority of people affected are children and adolescents [14,16,37,38]. Although we did not find significant associations between serology and epileptic seizures, as with psychosis, our findings therefore present a very high prevalence of epileptic seizures (both generalized and focal) that occur independently of seropositivity to cysticercosis and thus could be caused by other underlying diseases such as cerebral malaria, head trauma or perinatal hypoxia in children, to name a few.

When compared with other similar studies, our findings here corroborate with results from Bangoua, Cameroon [27], which did not find any statistically significant difference in the seropositivity to *T. solium* larva between people with epileptic seizures (1/61, 1.6%) and those without (5/323, 1.5%). Similarly, another study in the Democratic Republic of Congo [5] revealed that there was no difference in seropositivity between people with neurological disorders and those without. In addition, there was no association between cysticercosis and epilepsy in a study conducted in the Gambia [39]. To the contrary, it is well documented in the literature from Latin America, Asia and Africa, including the study done in Angónia district, that seropositivity to circulating antigens or antibodies of cysticercosis were associated with epilepsy in community-based studies [14,40,41,42]. Furthermore, the odds ratio from a study in Nigeria revealed that people with epilepsy in the study area with a relatively high cysticercosis prevalence, were two times more likely to be seropositive than those without [28].

In our study, chronic headache was significantly associated with circulating antigens (p = 0.013). In fact, other studies performed in Mozambique, Burkina Faso and Tanzania also

found that seropositivity to cysticercal antigens was associated with severe headache [14,15,42,43]. Nevertheless, in other settings such as in Anzoátegui in Venezuela, Vietnam and Nabo Village in Tiandong, China, no statistical association was found between these two variables [44–46]. Our findings may reflect the fact that headache may be correlated to the pathogenesis of the disease in this study setting.

In addition to the high burden of epileptic seizures and chronic headache, we also found that psychotic manifestations were highly prevalent in the setting of our study (18.3%), confirming the anecdotal reports from the Mocuba district community leaders of having a high proportion of community residents with those conditions. As with the other two disorders, psychotic disorders do not seem to be related to cysticercosis seropositivity either. Psychotic disorders are understudied in Africa, as revealed by a recent scoping review [47], that found only nine studies in which the prevalence of psychotic disorders was analysed in Africa from 1984 to 2020. The available reports describe a prevalence ranging from 1% to 4.4% in rural regions. The only published population-based study done in 2003 in a rural (Cuamba) and urban (Maputo) area of northern and southern Mozambique, respectively, found that seizure disorders (4.4% vs 1.6%), psychosis (4.4% vs 1.9%) and mental retardation (1.9% vs 1.3%) were higher in rural compared to urban areas [24] and well below the results of our study and recent hospital data from Zambézia Provincial Directorate [36]. It is difficult to explain this high burden. Apart from the possibility of other CNS pathogens causing epileptic seizures/epilepsy and/or psychosis, such as *Toxoplasma gondii*, *Toxocara spp*. and *Onchocerca volvulus* (the latter was recently identified in Zambézia province) [20,21,48,49], we think that the prolonged armed conflict that affected this region for decades in some of the study villages [50,51], combined with the stress it provokes in the affected population, might be playing a role as causative agent for this high prevalence of psychosis. Indeed, the effects of armed conflict on mental health within the affected communities is well documented in the literature, as it causes disruption of socio-economic and political infrastructure including the appropriate delivery of health care services, which was the case in our study setting and other regions of Mozambique [50–53].

### Risk factors and multivariate logistic regression

In our study, patients of older age were more likely to be seropositive for the parasite. Regarding age, the results are important because they will allow us to understand the transmission dynamics of TSC in the study region, and to identity the age groups at higher risk. It is likely that infection events accumulate as people age [54], and that *T. solium* cysticercus antibodies might persist for several years [55]. Similarly to our results, studies from Tanzania [43], Ecuador [56], Zambia [31], Peru [57], Burkina Faso [58] and Vietnam [29], including a systematic review [59], revealed an association between increasing age and seropositivity to cysticercosis, except for a study conducted by Noormahomed et al. [15], in which age was not significantly associated with the seroprevalence of the helminths studied, including cysticercosis.

Furthermore, in concordance with the present study, there was no statistically significant difference between gender and seropositivity in Ecuador [56] and Zambia [31]. To the contrary, in Tanzania [43], females were more likely to be seropositive, while in Burkina Faso and Vietnam this was true for males [29,58]. On the other hand, neither gender nor age presented a statistically significant association with seropositivity to cysticercosis in other studies [30, 44,46, 60].

Analysing the knowledge of *T. solium*, although our results did not present a statistically significant association with TSC seropositivity, in other countries such as Nigeria [28] poor knowledge of cysticercosis and improper pork preparation were identified as main risk and behavioural factors contributing to the high prevalence.

Other potential risk factors such as consumption of pork [61], pig keeping, use of unsafe water, faecal disposal and types of latrine were not statistically significant in our study, similarly to the study done in Angónia district [14]. This was expected since the study participants share the same environmental conditions, and are equally exposed to the same risk factors and consequently equally exposed to the infection. Furthermore, although there is a fraction of Muslim people amongst our study participants, we did not expect our findings to be affected by the religion, since cysticercosis is acquired mainly through water or food contaminated with *T. solium* eggs, and consequently anyone can acquire the infection regardless of the consumption of pork meat or professed religion. In some studies, food, water and hygiene related factors were significantly associated with cysticercosis seropositivity, including consumption of undercooked pork [44], and using the same water source [43]; although in other studies, factors such as consumption of raw meat and/or vegetables [30] and hygiene (presence of latrine) [57] were not associated with cysticercosis seropositivity.

## Strengths and limitations

The strength of this study is reflected in the fact that, to the best of our knowledge, this is the first assessment done in Mozambique aiming to define the seroprevalence of cysticercosis and its possible association with some selected neuropsychiatric disorders. It is therefore the first pilot community-based study representative of the Mocuba district population including rural and urban settings and all age groups except those under 2 years old. This baseline information will be useful to define strategies for control and mitigation of TSC in this district and to effectively diagnose and treat neuropsychiatric disorders that might be associated with cysticercosis. Additionally, further studies can be developed in other districts and provinces, taking into account the population at risk.

One of the limitations was the lack of clinical examination and diagnostic tools required to confirm the diagnosis of neuropsychiatric disorders. Also, patients with neuropsychiatric disorders with positive or negative serology for cysticercosis did not undergo CT or MRI examination in order to confirm or not the diagnosis. Regarding the serological assays, these present limitations in terms of sensitivity and specificity as well as on the discrimination of past or current infection. Additionally, serological screening for malaria, toxoplasmosis, toxocariasis, and onchocerciasis, that may cause neuropsychiatric disorders, was not done. Thus, it is not possible to draw firm conclusions about the fraction of participants whose neuropsychiatric signs/symptoms could have been due to cysticercosis/NCC or its sequelae.

## Conclusions and recommendations

In conclusion, our study found a seroprevalence of 10.3% of TSC and was associated with chronic headache (p = 0.013). Nonetheless, the seroprevalence was not associated with epileptic seizures/epilepsy, and/or psychosis. The lowest prevalence was found in Alto Benfica village (3.6%) and the highest in Mocuba Municipality (17.7%). It was also shown that seropositivity increased with age (p = 0.03). Our findings are important because there is a need to understand the burden of mental illness and its causes in the region, and whether it is related to cysticercosis/NCC.

Nevertheless, the role of TSC in the aetiology of neuropsychiatric disorders in cysticercosis endemic areas should be further studied. In addition, other CNS pathogens should be investigated to define their role in the burden of our targeted neuropsychiatric disorders as well as to help in the differential diagnosis of cysticercosis/NCC. This should be done in combination with studies of psychological, economic and societal impact of armed conflicts in the region and in sub-Saharan Africa as a whole.

Given the limited value of serology for diagnosis of cysticercosis and NCC and the scarcity and costliness of imaging techniques in endemic countries, research priorities should be given to the development of a point-of-care diagnostic tool and/or a biomarker. This should be done in combination with studies on parasite genetic diversity, and eco-epidemiological as well as clinical features of cysticercosis/NCC to assess geographic distribution patterns to support surveillance, diagnosis and control of cysticercosis and NCC.

## Acknowledgments

We are indebted to the Zambézia Provincial Governor, Dr Abdul Razak Noormahomed, his team and to the community leaders whose support and collaboration during the field work was critical for the engagement of the study participants in this research. We also thank the Mental health Department and the National Public Health Directorate of the Ministry of Health in Maputo, Mozambique for providing clinical assistance to the participants. Finally, we are thankful to the study participants who consented to participate in this study.

## Author Contributions

**Conceptualization:** Alberto Pondja, Clarissa Prazeres da Costa, Veronika Schmidt, Andrea S. Winkler, Emília Virgínia Noormahomed.

**Data curation:** Irene Langa, Fernando Padama, Noémia Nhancupe, Alberto Pondja, Dominik Stelzle, Emília Virgínia Noormahomed.

**Formal analysis:** Alberto Pondja, Dominik Stelzle, Clarissa Prazeres da Costa, Veronika Schmidt, Andrea S. Winkler, Emília Virgínia Noormahomed.

**Funding acquisition:** Andrea S. Winkler, Emília Virgínia Noormahomed.

**Investigation:** Irene Langa, Fernando Padama, Noémia Nhancupe, Alberto Pondja, Clarissa Prazeres da Costa, Veronika Schmidt, Andrea S. Winkler, Emília Virgínia Noormahomed.

**Methodology:** Irene Langa, Fernando Padama, Noémia Nhancupe, Emília Virgínia Noormahomed.

**Project administration:** Emília Virgínia Noormahomed.

**Supervision:** Irene Langa, Fernando Padama, Noémia Nhancupe, Emília Virgínia Noormahomed.

**Writing – original draft:** Irene Langa, Fernando Padama, Noémia Nhancupe, Alberto Pondja, Delfina Hlashwayo, Lidia Gouveia, Dominik Stelzle, Clarissa Prazeres da Costa, Veronika Schmidt, Andrea S. Winkler, Emília Virgínia Noormahomed.

**Writing – review & editing:** Irene Langa, Fernando Padama, Noémia Nhancupe, Alberto Pondja, Delfina Hlashwayo, Lidia Gouveia, Dominik Stelzle, Clarissa Prazeres da Costa, Veronika Schmidt, Andrea S. Winkler, Emília Virgínia Noormahomed.

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
