## [Decision Letter · Decision Letter 0]

24 Mar 2022

Dear Ms Hlashwayo,

Thank you very much for submitting your manuscript "The burden of T. solium cysticercosis and selected neuropsychiatric disorders in Mocuba district, Zambézia province, Mozambique" for consideration at PLOS Neglected Tropical Diseases. As with all papers reviewed by the journal, your manuscript was reviewed by members of the editorial board and by several independent reviewers. In light of the reviews (below this email), we would like to invite the resubmission of a significantly-revised version that takes into account the reviewers' comments. 

I have completed the evaluation of your manuscript based on comments from two reviewers who recommend reconsideration of your manuscript following major revision and modification. I invite you to resubmit your manuscript after addressing the comments below. 

We cannot make any decision about publication until we have seen the revised manuscript and your response to the reviewers' comments. Your revised manuscript is also likely to be sent to reviewers for further evaluation.

Sincerely,

Eduardo Torres

Guest Editor

Ricardo Fujiwara

Deputy Editor

Reviewer's Responses to Questions

**Key Review Criteria Required for Acceptance?**

**Methods**

-Are the objectives of the study clearly articulated with a clear testable hypothesis stated?

-Is the study design appropriate to address the stated objectives?

-Is the population clearly described and appropriate for the hypothesis being tested?

-Is the sample size sufficient to ensure adequate power to address the hypothesis being tested?

-Were correct statistical analysis used to support conclusions?

-Are there concerns about ethical or regulatory requirements being met?

Reviewer #1: - Line 49-50 in the abstract, should be corrected. 

- Line 177 - I would ask the authors to specify the age of those participants considered children and, thus, needed the consent of a guardian. They should also specify if assent was required. 

- The authors do not exclude those participants that recently moved into the area. They could have been infected in their previous residence. 

- Vignette 1-4: the authors should provide references. 

- Line 275-276: the authors do not adjust for having porks at home or in the neighbourhood. They should at least discuss how this could have affected in their results. They did not adjust for religion, what could have affected pork consumption.

Reviewer #2: The objectives of the study were clear enough and had a clear testable hypothesis. The study design generally addressed the outlined objectives, what may have been considered an experimental lapse was explained in the limitations section.

The sample size was representative enough to cover the hypothesis being tested. Appropriate statistical tools were used, however, the statistical analysis done for table 4 is incomplete. I have recommended that a bivariate analysis be first of all done between the serostatus of cysticercosis and the vignettes described in the study and the results presented clearly, before going ahead with the binary regression analysis.

Ethical permission/clearance was adequately sought and obtained.

**Results**

-Does the analysis presented match the analysis plan?

-Are the results clearly and completely presented?

-Are the figures (Tables, Images) of sufficient quality for clarity?

Reviewer #1: - Figure 2: English language should be reviewed. 

- Lines 293-295: Unclear if they used a statistical test to verify it. If it is not used, I recommend to use it. 

- Line 300-303: This statement is not correct.

Reviewer #2: The findings of the study and the analysis plan do corroborate/match one another. Generally, the results were clear enough to comprehend, however, some of the data presented need to be reconciled. The statistical analysis for Table 4 has to be revisited.

**Conclusions**

-Are the conclusions supported by the data presented?

-Are the limitations of analysis clearly described?

-Do the authors discuss how these data can be helpful to advance our understanding of the topic under study?

-Is public health relevance addressed?

Reviewer #1: Results should be interpreted with care. 

- Lines 321-323 should verify is this statement is true.

Reviewer #2: The conclusion needs to be rewritten to clearly convey the important information in the study. The major limitations of the study were mentioned by the authors. However, the authors did not discuss the public health relevance of the study.

**Editorial and Data Presentation Modifications?**

Reviewer #1: English language

Reviewer #2: I recommend that the manuscript be accepted subject to implementing the minor revisions recommended

**Summary and General Comments**

Reviewer #1: This is very relevant data for Mozambique, since there is not much data on T. solium presence in this country. However, I would like to see some clarifying edits and corrections.

Reviewer #2: This is a relevant baseline study which has generated important information on human cysticercosis in some parts of Mozambique. The information could be used to inform policy and practice and also serve as a reference for prospective studies. 

There was a flaw with some aspect of the experimental design that had to do with diagnosis of neuropsychotic disorders described in the studies, however, being a baseline community study, this could be overlooked, moreover, the authors actually mentioned this as a major limitation to their study.

PLOS authors have the option to publish the peer review history of their article (what does this mean?). If published, this will include your full peer review and any attached files.

Reviewer #1: No

Reviewer #2: No
---

## [Decision Letter · Decision Letter 1]

24 Jun 2022

Dear Ms Hlashwayo,

We are pleased to inform you that your manuscript 'The burden of T. solium cysticercosis and selected neuropsychiatric disorders in Mocuba district, Zambézia province, Mozambique' has been provisionally accepted for publication in PLOS Neglected Tropical Diseases.

The article was accepted, however check some observations and suggestions sent by reviewers after the second round of review and include them in the final version.

Best regards,

Eduardo Torres

Guest Editor

Ricardo Fujiwara

Deputy Editor

Reviewer's Responses to Questions

**Key Review Criteria Required for Acceptance?**

**Methods**

-Are the objectives of the study clearly articulated with a clear testable hypothesis stated?

-Is the study design appropriate to address the stated objectives?

-Is the population clearly described and appropriate for the hypothesis being tested?

-Is the sample size sufficient to ensure adequate power to address the hypothesis being tested?

-Were correct statistical analysis used to support conclusions?

-Are there concerns about ethical or regulatory requirements being met?

Reviewer #1: After my comment "- Line 275-276: the authors do not adjust for having porks at home or in the neighbourhood. They

should at least discuss how this could have affected in their results. They did not adjust for religion,

what could have affected pork consumption." I still believe that they should discuss that during the manuscript discussion. It is a very relevant point. The region under study has muslim population.

Reviewer #2: The objectives of the study were clearly articulated and adequately addressed with a good study design. The sample size was scientifically determined, sufficient enough to address the hypothesis tested and therefore acceptable.

The data derived from the study was analyzed using appropriate statistical analysis.

**Results**

-Does the analysis presented match the analysis plan?

-Are the results clearly and completely presented?

-Are the figures (Tables, Images) of sufficient quality for clarity?

Reviewer #1: After the authors comment "Thank you for this comment. We have now tested the difference in antibody

and antigen prevalence and the antibody prevalence is significantly higher

than the antigen prevalence (p<0.005). Also, the seroprevalence differs

significantly between districts (p=0.007). For both scenarios, Chi-Square tests

were used. We have added this in the manuscript in lines 326-331 (trackchanges).", I suggest to discuss this findings at the discussion of the manuscript.

Reviewer #2: The analysis presented matched the analysis plan. The results were clearly presented and captured using easy-to- understand tables and figures.

**Conclusions**

-Are the conclusions supported by the data presented?

-Are the limitations of analysis clearly described?

-Do the authors discuss how these data can be helpful to advance our understanding of the topic under study?

-Is public health relevance addressed?

Reviewer #1: (No Response)

Reviewer #2: The conclusions of the study supported the presented data. The authors also presented the limitations of the study and discussed its public health significance.

**Editorial and Data Presentation Modifications?**

Reviewer #1: (No Response)

Reviewer #2: I recommend that the paper be accepted.

**Summary and General Comments**

Reviewer #1: The authors have adress correctly the comments made during the revision. I still have a couple of small suggestions to improve the manuscript.

Reviewer #2: The authors have done a very thorough job effecting all major corrections/ recommendations made. However, I would like to draw their attention to my second comment in line 43 of the Abstract, referring to chronic headache not being a neuropsychiatric disorder, but a symptom.

Therefore, to adequately capture this, I suggest that the sentence be revised to read, "This study aimed to evaluate the association of the disease with selected neuropsychiatric disorders and/ or symptoms...."

PLOS authors have the option to publish the peer review history of their article (what does this mean?). If published, this will include your full peer review and any attached files.

Reviewer #1: No

Reviewer #2: No

---

## [Editor Report · Acceptance letter]

11 Jul 2022

Dear Ms Hlashwayo,

We are delighted to inform you that your manuscript, "The burden of *T. solium* cysticercosis and selected neuropsychiatric disorders in Mocuba district, Zambézia province, Mozambique," has been formally accepted for publication in PLOS Neglected Tropical Diseases.

Best regards,

Shaden Kamhawi

co-Editor-in-Chief

Paul Brindley

co-Editor-in-Chief
